# Visualisation tools for dependent peptide searches to support the exploration of *in vitro* protein modifications

**George W. Preston**[1,2¤]*, **Liping Yang**[2], **David H. Phillips**[1], **Claudia S. Maier**[2]*

**1** Department of Analytical, MRC-PHE Centre for Environment & Health, Environmental & Forensic Sciences, School of Population Health & Environmental Sciences, Faculty of Life Sciences & Medicine, King's College London, London, England, United Kingdom, **2** Department of Chemistry, Oregon State University, Corvallis, OR, United States of America

¤ Current address: Stoller Biomarker Discovery Centre, University of Manchester, Manchester, England, United Kingdom
* george.preston@kcl.ac.uk (GWP); claudia.maier@oregonstate.edu (CSM)

**Data Availability Statement:** All novel mass spectrometry data are available via PRIDE/ProteomeXchange (accession number PXD013040).

## Abstract

Dependent peptide searching is a method for discovering covalently-modified peptides–and therefore proteins–in mass-spectrometry-based proteomics experiments. Being more permissive than standard search methods, it has the potential to discover novel modifications (e.g., post-translational modifications occurring *in vivo*, or modifications introduced *in vitro*). However, few studies have explored dependent peptide search results in an untargeted way. In the present study, we sought to evaluate dependent peptide searching as a means of characterising proteins that have been modified *in vitro*. We generated a model data set by analysing *N*-ethylmaleimide-treated bovine serum albumin, and performed dependent peptide searches using the popular MaxQuant software. To facilitate interpretation of the search results (hundreds of dependent peptides), we developed a series of visualisation tools (R scripts). We used the tools to assess the diversity of putative modifications in the albumin, and to pinpoint hypothesised modifications. We went on to explore the tools' generality via analyses of public data from studies of rat and human proteomes. Of 19 expected sites of modification (one in rat cofilin-1 and 18 across six different human plasma proteins), eight were found and correctly localised. Apparently, some sites went undetected because chemical enrichment had depleted necessary analytes (potential 'base' peptides). Our results demonstrate (i) the ability of the tools to provide accurate and informative visualisations, and (ii) the usefulness of dependent peptide searching for characterising *in vitro* protein modifications. Our model data are available via PRIDE/ProteomeXchange (accession number PXD013040).

## Introduction

By the time a protein is subjected to analysis, it can have acquired one or more covalent modifications. These could include modifications of biological origin, modifications introduced

**Funding:** This work was supported by Cancer Research UK (https://www.cancerresearchuk.org; grant number CRUK/A14329 to D.H.P.) and King's College London (https://www.kcl.ac.uk). Mass spectrometry was conducted by the Oregon State University (OSU) Mass Spectrometry Center, supported in part by OSU's Research Office and institutional funds (https://oregonstate.edu). The procurement of the Orbitrap Fusion Lumos was made possible by a grant from the National Institutes of Health (https://www.nih.gov; grant number S10 OD020111 to C.S.M.). The funders had no role in study design, data collection and analysis, decision to publish, or preparation of the manuscript.

**Competing interests:** The authors have declared that no competing interests exist.

deliberately (e.g., to probe protein structure and function), and modifications occurring during sample preparation and storage. In bottom-up mass-spectrometry-based proteomics, where proteins are digested and analysed as peptides, prior knowledge of modifications can enable more of the acquired spectra to be identified [1]. Known or suspected modifications are specified as parameters of a database search, enabling more of the protein sequence to be mapped, and also allowing the modifications themselves to be localised and quantified. For partially characterised or unknown modifications, however, this approach is not practical: specifying a long list of variable modifications (e.g., as a way of capturing unknown modifications) would expand the database dramatically, lengthening the search duration and reducing the number of confidently identified spectra [2]. New types of search have been developed to address this problem [2–9]. They include 'open' database searches, which permit precursor ions with shifted masses [6]; and 'spectral pair' searches, in which unidentified spectra are matched to spectral libraries [5]. An example of the latter approach is dependent peptide (DP) searching [3]. In a typical case, a DP is a chromatographic feature that is not identified by a database search, but whose fragment-ion spectrum partially matches that of one of the search hits (the 'base' peptide). The DP is typically a modified form of the base peptide, and the two features' masses differ. In theory, some of the DP's product ions will be the same as the corresponding product ions of the base peptide, while others will display the mass difference ($\Delta m$). Crucially, $\Delta m$ does not need to be specified *a priori*, as it is calculated for every pair of unidentified feature and database-search hit. Identifying features in this way can take much less time than a database search [3] but does confer certain limitations: sites that are fully occupied by unknown modifications cannot be detected; and overall sequence coverage is unlikely to be extended.

Originally implemented as a stand-alone tool (ModifiComb [3, 10]), DP searching has recently been incorporated into the MaxQuant software [11, 12]. Within MaxQuant, the DP search can utilise hits (i.e., potential base peptides) generated by the Andromeda search engine [13]. Studies utilising MaxQuant's DP search function have confirmed its potential to discover modifications [12, 14–21]. Lassak et al. used the function to analyse a bacterial translation elongation factor, and discovered a novel type of glycosylation [14]. Mordret et al. used the function to detect single amino acid substitutions, and also carried out a general test of its validity [12]. A large set of synthetic phosphopeptides and corresponding unmodified peptides was analysed, and the phosphoryl modifications were left for a DP search to find. The search identified over a thousand spectra as belonging to singly-phosphorylated peptides, and all of these were true positives [12]. Few studies, however, have explored DP search results in an untargeted way. In the present study, we sought to evaluate DP searching as a means of characterising *in vitro* modified proteins. First, we generated model data by analysing a model protein (bovine serum albumin, BSA) that had been treated with a protein-modifying reagent (*N*-ethylmaleimide, NEM). Then, we performed DP searches and attempted to rationalise the search results.

Visualisation tools can greatly facilitate the interpretation of proteomic mass spectrometry data and database search results [22–25]. We identified a need for tools that visualise DP search results, and to meet this need we wrote a set of five scripts in the R language [26]. Three of the scripts are for surveying distributions of DPs (i.e., are hypothesis-generating), and the other two are for pinpointing hypothesised modifications (i.e., are hypothesis-testing). Some of the scripts can enrich DPs for modifications that are unique to a test sample. Herein we report search results and visualisations for our own data, as well as for public data from two other studies [27, 28]. The results demonstrate how a combination of DP searching and visualisation can assist in the characterisation of *in vitro* modified proteins. The approach could be useful for characterising protein targets of enzyme activities and reactive small molecules.

## Materials and methods

### Preparation and analysis of modified BSA

BSA (1 mg mL$^{-1}$) was reacted with NEM (1 mM) in potassium phosphate buffer (100 mM) at pH 7.4. Unreacted NEM was scavenged with 1,4-dithiothreitol (DTT). The protein was purified (buffer exchange), reduced (DTT), alkylated (iodoacetamide), purified again (acetone precipitation), and digested (trypsin). The peptides were analysed in duplicate (analytical replicates 1 and 2) by reversed-phase nano liquid chromatography (nanoACQUITY liquid chromatograph; Waters, Milford, Massachusetts, USA) with online data-dependent tandem mass spectrometry (Orbitrap Fusion Lumos mass spectrometer; ThermoFisher Scientific, Waltham, Massachusetts, USA). A control sample (untreated BSA) was also prepared and analysed. Further details of materials and methods can be found in S1 Text. Data for NEM-treated and untreated BSA have been deposited in PRIDE/ProteomeXchange [29] (accession number PXD013040).

### Public data

Further mass spectrometry data were obtained from PRIDE/ProteomeXchange [29]. Data were selected according to the following criteria: (i) experiment involving exposure of one or more proteins to a protein-modifying reagent; (ii) data collected using standard data-dependent acquisition mass spectrometry; (iii) control data and two replicates available (not an essential criterion); (iv) results of a variable-modification database search reported in the literature. The following files/groups of files met the criteria and were included in the present study (the selection was not exhaustive): DMF_Cofilin1A.raw from PRIDE project PXD008314 [27]; and 1362-cs774_0_a.raw, 1364-cs774_0_b.raw, 1380-cs774_5_a.raw and 1382-cs774_5_b.raw from PRIDE project PXD006663 [28]. Sequence data (*.fasta files) were obtained from the RCSB Protein Data Bank [30] (accession numbers 4F5S [31] and 1S81 [32]) and UniProt [33] (accession numbers P02042, P02647, P02766, P02768, P02787, P45592 and P68871, and the UniProtKB/Swiss-Prot human proteome, 4$^{th}$ June 2019). Where possible, sequences were obtained without extraneous elements such as signal peptides. The data from Protein Data Bank accession number 4F5S consisted of two identical sequences, and so one of these was removed. MaxQuant's database of contaminants [11] was used either as supplied or in an edited form (see S1 Text).

### Database searches and dependent peptide searches

All searches were done in MaxQuant (Max Planck Institute of Biochemistry, version 1.6.0.1) [11]. Database searches were done using Andromeda [13]. Individual *.raw files were searched against databases consisting of either a protein of interest plus potential contaminants (1 + 244 or 1 + 245 sequences) or the UniProtKB/Swiss-Prot human proteome plus potential contaminants (20,406 + 82 sequences). The *in silico* digestion was done in 'specific' mode, using 'Trypsin/P' as the enzyme, and allowing for a maximum of two missed cleavages. The maximum peptide mass was adjusted so as to include $\geq$95% of relevant theoretical peptides (see S1 Text). The minimum peptide length was seven amino acid residues. Additional *in silico* digestions were done using PeptideMass [34] (see S1 Text).

When the purpose of the database search was to discover potential base peptides, a minimal set of variable modifications (methionine oxidation and protein N-terminal acetylation) and an appropriate fixed modification (cysteine *S*-carbamidomethylation [28] or *S*-pyridylethylation [27]) were specified. When the purpose was instead to localise a particular modification (a variable-modification search), the fixed and variable modifications were adjusted accordingly

(see S1 Text). The maximum number of modifications per peptide was always five, and the 'second peptides' function was always enabled. DP searches were appended to their respective database searches by enabling the 'dependent peptides' function (false discovery rate of 1%, mass bin size of 0.0065 Da). Results of DP searches were obtained from allPeptides.txt files [11] and filtered as described below. For selected DPs, we investigated whether the same chromatographic feature could also be detected by a variable-modification search (see S1 Text).

### Development of visualisation tools

All tools (S1–S5 Scripts) were developed in R for Windows (R Core Team, version 3.4.0 or later) [26] using functions from the base packages, plus the additional function *read.fasta* from the 'seqinR' package (version 3.3–6 or later) [35]. S1–S4 Scripts were developed and tested on a Dell desktop PC (Intel Core i5-7500 processor, 8 GB RAM) running Windows 10. S5 Script was developed and tested on a Toshiba laptop PC (AMD E1-2100 APU processor, 4 GB RAM) running Windows 8.1. Each script requires a set of search results (allPeptides.txt files), a protein sequence (*.fasta file), and the identifier of a protein of interest (e.g., a UniProt identifier). The search results are filtered (see S1 Text of Table B) and a table of DPs is prepared. DPs are localised to segments of the protein sequence using a 'sliding window' [36]. Different scripts require different numbers of allPeptides.txt files and process the data to different extents. S1, S2 and S3 Scripts are for surveying DPs' $\Delta m$ values; they return DP localisation plots and $\Delta m$ frequency histograms. S4 and S5 Scripts are for pinpointing particular modifications; they require an expected $\Delta m$ value, and they return plots of DPs' localisation probabilities [12]. S2, S3 and S4 Scripts have the ability to enrich the DPs for modifications that are unique to a test sample (see 'Results and discussion'). A set of notes explaining how the scripts work can be found in S1 Text of Table C. We will endeavour to maintain the scripts' compatibility with relevant software, and to address any limitations that come to light. Any future versions of the scripts will be made available from https://github.com/preston-gw/.

The accuracy of the visualisations was tested by manually mapping selected DPs onto graphics generated by scripts (20 DPs across five graphics, all confirmed to have been mapped correctly). After this, no significant changes were made to either the main data processing code or the mapping code (changes to graphical parameters, for example, were permitted). Certain graphics were prepared in batches by iterating an appropriate script. Figures were prepared from R output files using Inkscape (Free Software Foundation, version 0.91 or later) and GNU Image Manipulation Program (GIMP Development Team, version 2.10.8) (see S1 Text). Figures such as S4 Fig and S8 Fig are representative of the graphics generated by the scripts.

### Visualisation tools' instructions for use

1. Open R (version 3.6.0 or later)

2. Make sure that package 'seqinR' [35] is installed. Installation is achieved by entering *install. packages("seqinr")* in the R console.

3. Open the script within R (File > Open script)

4. Review the script and complete file paths as appropriate. Further instructions are included in the header and body of the script. Explanatory notes can be found in S1 Text of Tables B and C.

5. Save a copy of the script if desired (File > Save as)

6. Run the script (Edit > Run all)

7. A graphic should appear on-screen. The times taken to visualise our model data using one of the aforementioned PCs were as follows: S1 Script, 8 s (desktop); S2 Script, 27 s (desktop), S3 Script, 62 s (desktop); S4 Script, 16 s (desktop); S5 Script, 26 s (laptop).

## Mass calculations

Expected $\Delta m$ values were calculated in R (version 3.4.0 or later) [26] using monoisotopic masses from ChemDraw (various versions, PerkinElmer) or Unimod [37] (http://www.unimod.org/masses.html). Masses used for calculations were accurate to at least four decimal places. $\Delta m$ values mentioned in the text have been rounded to two decimal places.

## Statistical methods

Pairwise comparisons of $\Delta m$ frequency histograms were done using the *cor.test* function (Spearman method) in R (version 3.4.0) [26].

## Results and discussion

### Exploration of model data

BSA was selected as a model protein because it is well-characterised and contains a number of nucleophilic (i.e., potentially modifiable) amino acid residues [38, 39]. NEM was selected as the protein-modifying reagent because it is reactive towards a variety of amino acid side chains (those of cysteine, lysine and histidine) [40]. We predicted that NEM would modify BSA's only reduced cysteine residue, Cys-34 [31], as well as one or more lysine and/or histidine residues. The primary products of the reaction of BSA with NEM were expected to be Michael adducts, in which a hydrogen atom of the protein has effectively been replaced by an *N*-ethyl-succinimidyl (NESyl) group ($\Delta m$ = +125.05 Da). NESyl groups attached to cysteine residues are susceptible to hydrolysis (additional $\Delta m$ = +18.01 Da) [41, 42], and we assumed that this would also be the case for NESyl groups attached to lysine or histidine residues. Additionally, sulfur atoms to which NESyl groups are attached may oxidise [41].

Large numbers of chromatographic features were detected in analyses of NEM-treated BSA ($N \geq 34{,}812$), and also in analyses of untreated BSA ($N \geq 44{,}430$). Five to six percent of the features ($1958 \leq N \leq 2328$) were identified by MaxQuant as either potential base peptides (Andromeda search hits, 23–25% of identified features) or DPs (75–77% of identified features) (S1 Fig). Peptides of BSA (95% of identified features) were 9–10 times as numerous as predicted by *in silico* digestion and oxidation ($N = 218$). The high ratio of observed to expected features implies that large numbers of modifications had occurred independently of NEM treatment (e.g., artefacts of sample preparation or modifications pre-existing in the BSA). The detection of so many 'background' modifications, although difficult to account for, is consistent with Nielsen and coauthors' estimate of 8–12 modified peptides per unmodified tryptic peptide [10]. Filters were employed to isolate the DPs, to limit the number of 'background' modifications (see S1 Text of Table B), and to limit $\Delta m$ to ±500 Da (for clarity of visualisation). Filtering removed 70–74% of the identified features (S1 Fig).

### Visualisation of $\Delta m$ distributions

The filtered $\Delta m$ values were visualised in two ways: firstly by mapping DPs to segments of the protein sequence, and secondly using a frequency histogram [3, 25]. These modes of visualisation, both achieved using S1 Script, revealed a diversity of putative modifications in NEM-

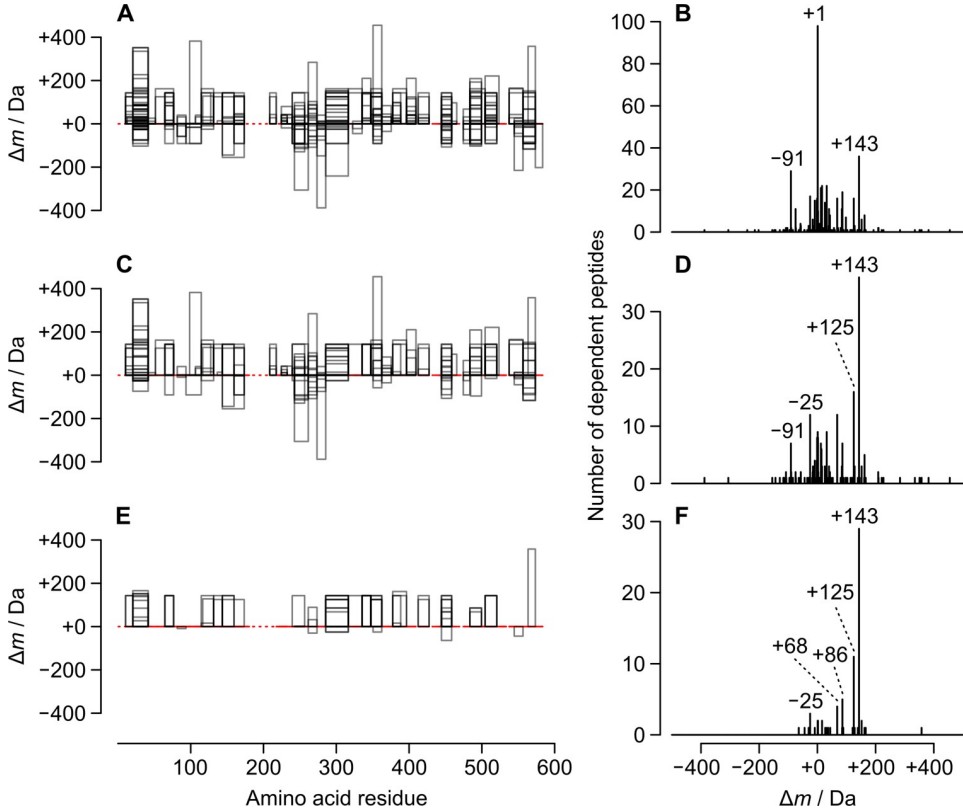

**Fig 1. Localisation plots and mass-shift (Δ*m*) frequency histograms for dependent peptides (DPs) of *N*-ethylmaleimide-treated bovine serum albumin.** In the localisation plots (left-hand panels), the protein sequence is represented as a dashed line that becomes solid in regions for which peptides were observed. X-axis values refer to positions in the protein sequence (position 1 = N-terminal amino acid residue). Each DP is represented as a rectangle whose height is proportional to Δ*m*, and whose grey border is partially transparent. The Δ*m* values are summarised in frequency histograms (right-hand panels). The DPs were unenriched (A, B), enriched using S2 Script (C, D) or enriched using S3 Script (E, F).

treated BSA (Fig 1A and 1B). Visualisations of putative modifications to porcine trypsin, also detected in analyses of NEM-treated BSA, demonstrated the flexibility of S1 Script (S2 Fig).

Δ*m* frequency histograms were used to investigate whether modifications in NEM-treated and untreated BSA were the same. Histograms for the two samples were similar (nearly as similar, or more similar, than histograms for analytical replicates; see S3 Fig). This suggested that the samples had many modifications in common. In order to selectively visualise the NEM-derived modifications, we investigated ways of enriching DPs. We started with a method (S2 Script) that subtracts the DPs observed in an analysis of untreated protein from those observed in an analysis of treated protein (Fig 1C and 1D). For this purpose, DPs were regarded as simple combinations of sequence and Δ*m* (in principle, retention time could also be used, but this was not attempted here). We then took the enrichment idea a stage further (S3 Script) by looking for DPs that were 'constantly conjoined' with NEM treatment (i.e., observed in both analyses of NEM-treated BSA, but in neither analysis of untreated BSA). Enrichment was quantified as an increase in the percentage of DPs having either of two putative NEM-derived groups: intact NESyl (Δ*m* ± tolerance = +125.05 ± 0.01 Da) or hydrolysed NESyl (Δ*m* ± tolerance = +143.06 ± 0.01 Da). Both modifications were observed for NEM-treated BSA (8.7% of the DPs from analysis 1, Fig 1B) and neither was observed for untreated BSA. S2 Script effected 2.3-fold enrichment of DPs from analysis 1 of NEM-treated BSA (Fig 1D). S3 Script effected

6.0-fold enrichment of these same DPs, but its greater stringency led to the exclusion of eleven relevant DPs (Fig 1F). The results suggest that our scripts should be able to enrich DPs even when the modifications of interest are unknown. In other words, the scripts might be able to discover novel modifications and attribute them to a given reagent or condition.

## Visualisation of expected modifications

As well as surveying the diversity of modifications, we were also interested in visualising the distributions of specific hypothesised modifications. For this purpose, we developed a method that maps 'constantly conjoined' DPs to the protein sequence and highlights their probable sites of modification (S4 Script). A sliding window is used as before, but in S4 Script its role is to direct the entry of values into matrices. Each DP is 'etched' (as a line of ones) into a blank matrix, and the localisation probabilities for that DP's modification are deposited in a corresponding zero matrix (localisation probability is a computed measure of the likelihood of a modification occurring at a given site [11, 12]). The two matrices are subsequently converted to images (R function *image* [26]) and merged. Fig 2 is a formatted version of one of the resulting graphics, showing the distribution of putative hydrolysed NESyl groups. S4 Fig is an example of an unformatted graphic, showing the distribution of putative intact NESyl groups. Note how some of the DPs in Fig 2 occur as pairs of putative diastereoisomers [43] (i.e., modified peptides with identical *m/z* values and fragmentation patterns but different retention times).

Using the hydrolysed NESyl group as an example, we examined whether the putative modifications had been localised to plausible sites. In 23 of 34 DPs with putative hydrolysed NESyl groups (68%), the highest localisation probability had been assigned to a cysteine, histidine or lysine residue (S1 Table). In cases where the same probability had been assigned to multiple sites, we used a system of prioritisation to narrow down the possibilities (see S1 Text for details). Six of the 23 plausible localisations were confirmed by a variable-modification search (S1 Table). In all six cases, the modification could be localised to a histidine or lysine residue. Modifications to cysteine residues, which represent a special case, are discussed below. A seventh, less plausible localisation (to Asp-13) was also confirmed by the variable-modification search (S1 Table).

In order to discover the DPs with modified cysteine residues, we had to account for the fact that the corresponding residues in the base peptides would also be modified (*S*-carbamido-methylated). Subtracting the $\Delta m$ for carbamidomethylation gave new values for NESyl (+68.03 Da) and hydrolysed NESyl (+86.04 Da), both of which we recognised from the $\Delta m$ frequency histograms (Fig 1). Surprisingly, neither modification was localised to Cys-34 (S5 and S6 Figs), and no modified cysteine residues were confirmed by the variable-modification search. Seeking to understand the apparent absence of modifications to Cys-34, we turned to a group of unexplained DPs ($\Delta m = -25.03$ Da; Fig 1F), which we speculated might contain oxidised cysteine residues (cysteinesulfinic acid). The modification was found to have been localised to Cys-34 in some DPs, but none that were 'constantly conjoined' with NEM treatment (S7 Fig). The ambiguous results for Cys-34 are possibly a consequence of modifications at this site having decomposed prior to or during analysis. We did see some evidence of modification to cysteine residues other than Cys-34 (S1 Table, S5–S7 Figs), and this was unexpected because these residues are normally disulfide-bonded to other cysteine residues [31]. It is possible that modifications to cysteine residues other than Cys-34 occurred when DTT was added to scavenge unreacted NEM. It is also possible that some of the other modifications observed in the study occurred following this addition of DTT.

The above results highlight the fact that observed $\Delta m$ values do not always correspond to real chemical transformations, and cannot always be interpreted directly. Direct interpretation

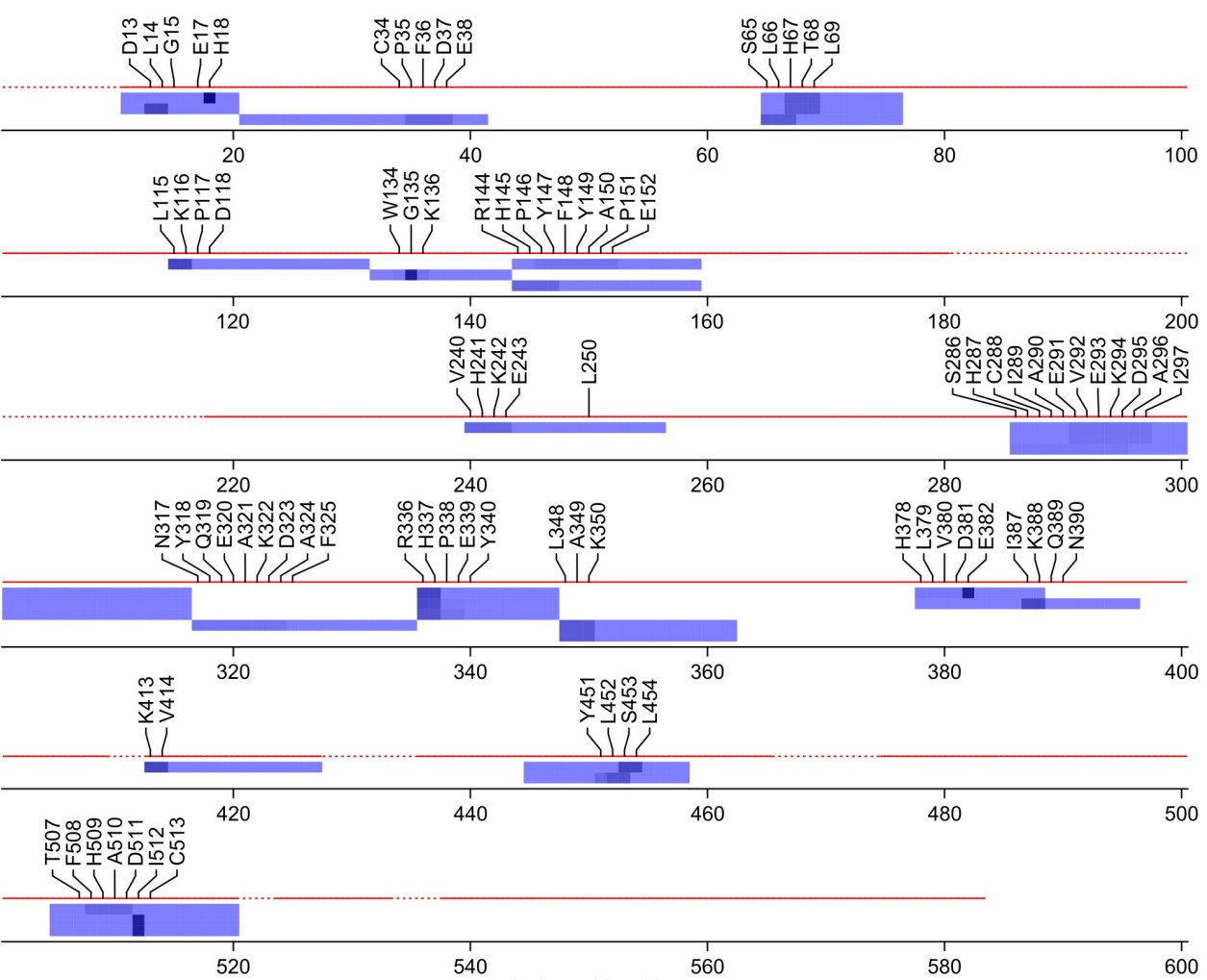

**Fig 2. Localisation plot for putative hydrolysed *N*-ethylsuccinimidyl groups in *N*-ethylmaleimide-treated bovine serum albumin (Δ*m* ± tolerance = +143.06 ± 0.01 Da).** The protein sequence is represented as a dashed line that becomes solid in regions for which peptides were observed. X-axis values refer to positions in the protein sequence (position 1 = N-terminal amino acid residue). Dependent peptides are represented as coloured strips with shading to indicate the localisation probability (darker = more probable). Any site with a non-zero probability is annotated. One dependent peptide (amino acid residues 336–347, probably modified at Arg-336 or His-337) does not appear because the relevant matrices were full.

is permitted if the DP is singly-modified and the base peptide is truly unmodified, and likewise if the DP and base peptide contain modifications that 'cancel out' (e.g., cysteine *S*-carbamido-methylation). However, if the base peptide contains modifications not found in the DP, or if the base peptide and DP contain different modifications at the same site, then interpretation will be less straightforward. Problems of this nature could be avoided by omitting modifications from the database search, but this would of course restrict the overall number of identifications.

## Validation of tools using public data

We explored the scripts' generality by applying them to analyses of public data. We hypothe-sised that a combination of DP searching and visualisation would reveal the same adducts as other authors had found by variable-modification database searching. We expected to observe

these adducts via targeted visualisation (e.g., using S4 Script). We also predicted that they would be evident from an untargeted survey (e.g., using S3 Script).

First, we analysed data from a study by Piroli et al. [27]. In this work, cultured cells (rat astrocytes) were exposed to the drug dimethyl fumarate. The authors collected proteins from the exposed cells, resolved them on gels, and then subjected individual protein bands to reduction, alkylation (4-vinylpyridine), digestion (trypsin) and analysis. In one of the protein bands, the authors detected a monomethyl fumarate adduct of cofilin-1 and localised the modification to a cysteine residue (Cys-139). Using the methods developed for the analysis of NEM-treated BSA, we performed an independent analysis of Piroli and coauthors' cofilin-1 data. S1 Script was used to survey $\Delta m$ values, and an additional script was developed for mapping the expected $\Delta m$ (S5 Script, an analogue of S4 Script that works with single allPeptides.txt files). S1 Script did not highlight the expected nominal $\Delta m$ (+25 Da; S8 Fig), which is perhaps understandable given this script's inability to enrich DPs. The overall sparsity of S8 Fig could reflect a real lack of modifications in the rat cofilin-1, or alternatively it could reflect qualities of the sample and/or data. S5 Script revealed that the expected $\Delta m$ (+-24.97 ± 0.01 Da) was present in one DP, and that it had been localised to the correct cysteine residue (S9 Fig). The $\Delta m$ itself is also evidence of correct localisation, since this is the difference in mass with respect to a pyridylethylated base peptide. The discrepancy in the site numbers (138 versus 139) arose probably because we used the sequence of mature rat cofilin-1 (no N-terminal methionine residue) whereas Piroli et al. used the sequence of the full-length protein (we used mature sequences where possible to ensure that N-terminal peptides could be found by Andromeda).

Further data were from a study by Salomón et al. [28]. In this work, the reactivity of human blood proteins towards a metabolite, 3-hydroxy-2,5-hexanedione, was explored using an alkyne-functionalised probe ('alk-3-HHD'). The authors prepared plasma from probe-treated blood and collected the plasma proteins. The proteins were reduced, alkylated (iodoacetamide) and digested (trypsin), and the resulting peptides were chemically enriched for alk-3-HHD adducts. The authors detected two different types of modification ('HTO' and 'HDMP', both specific to lysine residues) among six polypeptides (apolipoprotein A-I, haemoglobin β- and δ-chains, serotransferrin, serum albumin, and transthyretin). In total, the authors detected 18 unique sites of modification among 22 analytes. Again, we applied methods that had been developed for the BSA adducts. Salomón and coauthors' dataset included replicates and a control, permitting the use of S3 and S4 Scripts. To maximise contrast, we used data for the highest concentration of alk-3-HHD. S3 Script revealed multiple 'constantly conjoined' DPs of each of the six aforementioned polypeptides, but did not highlight any of the expected $\Delta m$ values (Fig 3A and 3B; S10–S14 Figs). Six DPs mapped to both of the haemoglobin chains, and therefore could not be localised unambiguously. S4 Script was used to map HTO- and HDMP-type modifications (with or without sulfonation [28]) to the sequences of the polypeptides. In total, 14 DPs with putative alk-3-HHD-derived modifications were detected (11 unique combinations of sequence and modification) (S2 Table). In each of the DPs, a lysine residue was either the site with the highest localisation probability, or was one of multiple such sites. Most of the DPs (93%) were of either haemoglobin β-chain (Fig 3C) or serum albumin (S15 Fig). Of the 22 analytes reported by Salomón et al., we detected six (27%) as DPs. Of the 18 expected sites of modification, we observed seven (39%). It is perhaps unsurprising that some of the expected DPs were not detected, since the chemical enrichment performed by Salomón et al. had the potential to remove their corresponding base peptides (an effect alluded to by Tyanova et al. in their protocol [11]). Indeed, for seven of the 22 expected analytes (32%), the absence of a required base peptide was sufficient to explain the absence of the DP.

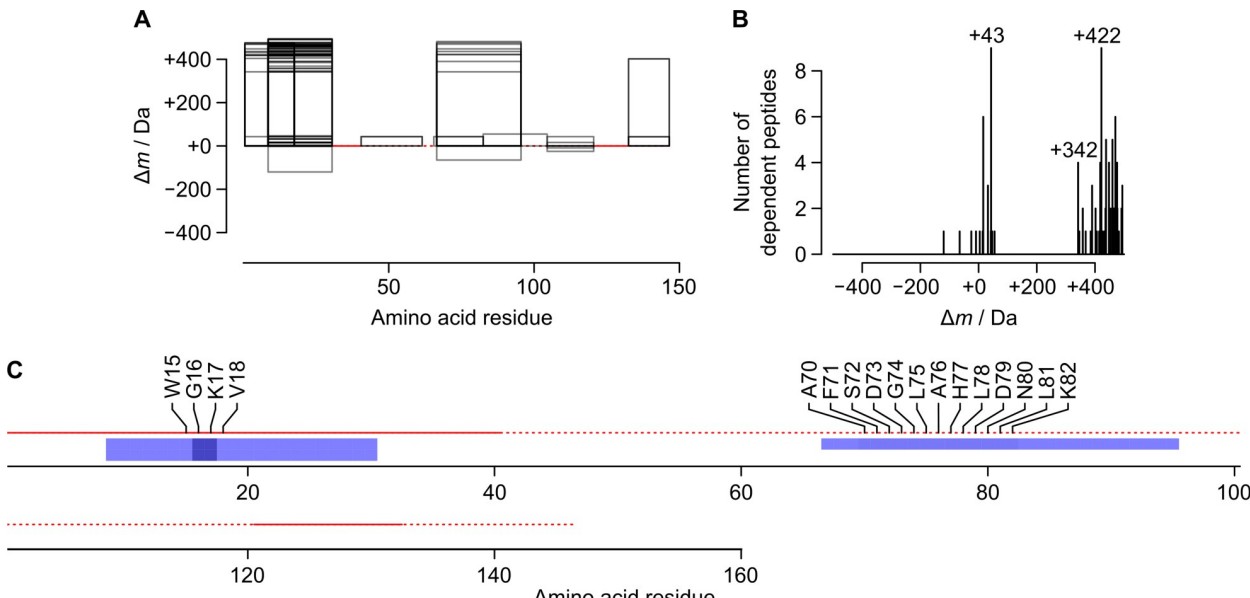

**Fig 3. Visualisation of dependent peptides of human haemoglobin β-chain using public data from the study by Salomón et al. [28].** (A) Dependent peptide localisation plot. (B) Mass-shift ($\Delta m$) frequency histogram. (C) Localisation plot for putative sulfonated 'HDMP'-type modifications ($\Delta m \pm$ tolerance = +390.10 ± 0.01 Da) [28]. Six of 106 dependent peptides also mapped to the sequence of haemoglobin δ-chain (S11 Fig). Base peptide VLGAFSDGLAHLDNLKGTFATLSELHCDK went undetected in analyses of untreated proteins and therefore does not appear (see S1 Text of Table C).

## Scope for extending the present study

There is scope beyond the present study for developing and integrating the visualisation tools. As R scripts, they are highly amenable to modification, and could be adapted for more specialised purposes. The filters and the $\Delta m$ tolerance could made more stringent or permissive as required. The plots could be customised by changing the colour scheme or narrowing the limits of the $\Delta m$ axes. Another idea would be to rotate the histogram so as to align its $\Delta m$ axis with that of the DP localisation plot.

One area in which there is significant scope for development is annotation. Currently, the scripts highlight mass shifts but do not attempt to identify them. Some identifications are already made by MaxQuant, and these could be easily transferred onto the plots. Further identifications could be made via public protein-modification databases such as Unimod [37] and RESID [44]. These databases contain calculated $\Delta m$ values via which observed $\Delta m$ values could be linked to modifications' identities. Identities could be added to the plots or visualised separately (e.g., a word-cloud of modifications' names). Another idea would be to highlight particular features of the protein sequence, such as nucleophilic amino acid residues. This could be done using lines, symbols or text.

Currently, each script visualises results for a single protein (S1–S3 Scripts) or combination of protein and $\Delta m$ (S4 and S5 Scripts). One way of extending the approach would be to iterate scripts so that they cycle through lists of proteins and/or $\Delta m$ values (in fact, we prepared certain groups of figures in this way). In theory, this could be done in a 'data-dependent' fashion by extracting the lists directly from allPeptides.txt. If this were attempted, each graphic (e.g., *.svg file) would have to be stamped with the protein identifier and/or $\Delta m$ value.

S2, S3 and S4 Scripts are able to enrich DPs for modifications that are unique to a test sample. The modes of enrichment employed by the scripts are simple but should work well for many *in vitro* modifications (especially modifications not found *in vivo*). The visualisation of

*in vivo* modifications would be an impactful next step, but one that might require a more quantitative approach: it would be helpful to visualise differences in abundance in addition to the simple difference between presence and absence.

Finally, there is scope for combining the scripts in an R package [45]. This would promote their usability beyond the present study.

## Summary

We have developed and tested a set of analytical tools with which to interpret the results of a DP search. The tools visualise putative modifications ($\Delta m$ values) for a protein of interest (either an isolated protein or a component of a proteome). Some of the tools are able to enrich DPs for modifications that are unique to a test sample. We envisaged that an untargeted survey of DPs (using S1–S3 Scripts) might generate hypotheses that could then be tested via targeted visualisation (using S4 and S5 Scripts). This approach helped us to achieve our aim of rationalising DP search results for NEM-treated BSA. Expected modifications were found, and the majority of these were localised to chemically plausible sites. In formal tests involving public data, a number of expected modifications were detected and correctly localised (although here the methods for surveying $\Delta m$ values proved less helpful than they had for the BSA study). Analyses of cysteine-specific modifications led us to consider the effect of fixed and variable modifications on $\Delta m$; and analyses of data for chemically enriched peptides led us to consider the potential of chemical enrichment to limit DP searches. We conclude (i) that the tools can summarise DP search results accurately and informatively, and (ii) that DP searching can be useful for characterising *in vitro* modified proteins.

## Supporting information

**S1 Fig. Numbers of features identified in analyses of NEM-treated and untreated BSA.** After filtering, the numbers of dependent peptides (DPs) were all similar (no two counts differed by more than 5%). 'Conjoined' DPs were those detected in analysis 1 of NEM-treated BSA and not detected in analysis 1 of untreated BSA. 'Constant' DPs were those detected in both analyses of NEM-treated BSA. 'Constantly conjoined' DPs were those detected in both analyses of NEM-treated BSA and not detected in either analysis of untreated BSA. (TIF)

**S2 Fig. Localisation plot and mass-shift frequency histogram for dependent peptides of porcine trypsin.** MaxQuant identified three putative deamidations and a putative methylation. Two of the deamidations were localised to asparagine residues. (TIF)

**S3 Fig. Similarity of mass-shift frequency histograms.** Three pairwise comparisons are shown: treated/treated, untreated/untreated and treated/untreated. The treated/untreated pair shown is the least similar of four possible combinations. $\rho$ = Spearman correlation coefficient. (TIF)

**S4 Fig. Probability localisation plot for putative intact NESyl groups in NEM-treated BSA ($\Delta m$ ± tolerance = +125.05 ± 0.01 Da).** X-axis values refer to positions in the protein sequence. (TIF)

**S5 Fig. Probability localisation plot for additional intact NESyl groups in NEM-treated BSA ($\Delta m$ ± tolerance = +68.03 ± 0.01 Da).** X-axis values refer to positions in the protein

sequence.
(TIF)

**S6 Fig. Probability localisation plot for additional putative hydrolysed NESyl groups in NEM-treated BSA ($\Delta m$ ± tolerance = +86.04 ± 0.01 Da).** X-axis values refer to positions in the protein sequence.
(TIF)

**S7 Fig. Probability localisation plot for putative oxidations in NEM-treated BSA ($\Delta m$ ± tolerance = −25.03 ± 0.01 Da).** X-axis values refer to positions in the protein sequence.
(TIF)

**S8 Fig. Localisation plot and mass-shift frequency histogram for dependent peptides of cofilin-1 from dimethyl-fumarate-treated rat astrocytes [27].** The nominal $\Delta m$ of +105 Da is consistent with pyridylethylation of non-cysteine residues.
(TIF)

**S9 Fig. Probability localisation plot for 1-carboxy-2-methylcarboxyethyl groups in cofilin-1 from dimethyl-fumarate-treated rat astrocytes [27] ($\Delta m$ ± tolerance = +24.97 ± 0.01 Da).** X-axis values refer to positions in the protein sequence.
(TIF)

**S10 Fig. Localisation plot and mass-shift frequency histogram for dependent peptides of apolipoprotein A-I from alk-3-HHD-treated human blood [28].**
(TIF)

**S11 Fig. Localisation plot and mass-shift frequency histogram for dependent peptides of haemoglobin δ-chain from alk-3-HHD-treated human blood [28].** All six dependent peptides also mapped to the sequence of haemoglobin β-chain (Fig 3).
(TIF)

**S12 Fig. Localisation plot and mass-shift frequency histogram for dependent peptides of serotransferrin from alk-3-HHD-treated human blood [28].**
(TIF)

**S13 Fig. Localisation plot and mass-shift frequency histogram for dependent peptides of serum albumin from alk-3-HHD-treated human blood [28].**
(TIF)

**S14 Fig. Localisation plot and mass-shift frequency histogram for dependent peptides of transthyretin from alk-3-HHD-treated human blood [28].**
(TIF)

**S15 Fig. Probability localisation plot for putative sulfonated HDMP-type modifications in serum albumin from alk-3-HHD-treated human blood [28] ($\Delta m$ ± tolerance = +-390.10 ± 0.01 Da).** X-axis values refer to positions in the protein sequence.
(TIF)

**S1 Script. An R script that filters dependent peptides and generates a dependent-peptide localisation plot and a mass-shift frequency histogram.**
(R)

**S2 Script. An R script that filters dependent peptides, enriches them on the basis of 'conjunction' and generates a dependent-peptide localisation plot and a mass-shift frequency**

**histogram.**
(R)

**S3 Script. An R script that filters dependent peptides, enriches them on the basis of 'constant conjunction' and generates a dependent-peptide localisation plot and a mass-shift frequency histogram.**
(R)

**S4 Script. An R script that isolates dependent peptides with a specified mass shift, enriches them on the basis of 'constant conjunction' and generates a probability localisation plot.**
(R)

**S5 Script. An R script that isolates dependent peptides with a specified mass shift and generates a probability localisation plot.**
(R)

**S1 Table. Dependent peptides and matching variable-modification search results.** Localisation probabilities for hydrolysed NESyl groups are given in parentheses after the respective amino acid symbols. Potential sites of modification are underlined, with the most plausible sites in boldface (see S1 Text). Dependent peptides were allowed to have either of two mass shifts: +86.04 ± 0.01 Da or +143.06 ± 0.01 Da (see 'Results and discussion').
(XLSX)

**S2 Table. Dependent peptides with putative alk-3-HHD-derived modifications.** Dependent peptides containing HDMP-/HTO-type modifications were identified using S4 Script. All modifications were detected as sulfonyl derivatives, and all could be localised to lysine residues. DPs were matched to Salomón and coauthors' search results [28] ('+' = match) by sequence, site of modification and modification type ('Expected analyte'), or by protein site only ('Expected site'). Two site numbers are given: the number used for the matching (first number); and the equivalent number for the mature protein (second number, in parentheses).
(XLSX)

**S1 Text. Supplementary methods.** Chemicals; Preparation of BSA adducts; Sample preparation for mass spectrometry; Nano liquid chromatography and mass spectrometry; Enumeration of tryptic peptides; Calculation of maximum peptide mass; Contaminant databases; Comparison of dependent-peptide and variable-modification search results;
Figure preparation; References; Table A (Gradient elution timetable); Table B (Criteria used to filter dependent-peptide search results); Table C (Explanatory notes to accompany scripts).
(PDF)

## Author Contributions

**Conceptualization:** George W. Preston, David H. Phillips, Claudia S. Maier.

**Data curation:** George W. Preston.

**Formal analysis:** George W. Preston.

**Funding acquisition:** David H. Phillips, Claudia S. Maier.

**Investigation:** George W. Preston, Liping Yang.

**Methodology:** George W. Preston, Liping Yang, Claudia S. Maier.

**Resources:** Liping Yang, Claudia S. Maier.

**Software:** George W. Preston.

**Supervision:** David H. Phillips, Claudia S. Maier.

**Validation:** George W. Preston.

**Visualization:** George W. Preston.

**Writing – original draft:** George W. Preston.

**Writing – review & editing:** George W. Preston, Liping Yang, David H. Phillips, Claudia S. Maier.

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
