## [Decision Letter · Decision Letter 0]

5 Mar 2020

PONE-D-20-02144

Visualisation tools for dependent peptide searches to support the exploration of *in vitro* protein modifications

PLOS ONE

Dear Dr Preston,

Thank you for submitting your manuscript to PLOS ONE. After careful consideration, we feel that it has merit but does not fully meet PLOS ONE’s publication criteria as it currently stands. Therefore, we invite you to submit a revised version of the manuscript that addresses the points raised during the review process.

As shown by the numerous questions of the reviewers the manuscript is currently not clear enough on the actual use of the visualisation tools that are proposed. The lack of details in the case studies and the subsequent confusion in understanding the visualisation is a concern. The revised version should not only include more detailed information but also refer to more of the existing resources describing PTMs. 

We would appreciate receiving your revised manuscript by April 27, 2020. To enhance the reproducibility of your results, we recommend that if applicable you deposit your laboratory protocols in protocols.io, where a protocol can be assigned its own identifier (DOI) such that it can be cited independently in the future. For instructions see: http://journals.plos.org/plosone/s/submission-guidelines#loc-laboratory-protocols

We look forward to receiving your revised manuscript.

Kind regards,

Frederique Lisacek

Academic Editor

PLOS ONE

Journal Requirements:

Reviewers' comments:

Reviewer's Responses to Questions

**Comments to the Author**

1. Is the manuscript technically sound, and do the data support the conclusions?

Reviewer #1: Partly

Reviewer #2: Yes

2. Has the statistical analysis been performed appropriately and rigorously? 

Reviewer #1: Yes

Reviewer #2: N/A

3. Have the authors made all data underlying the findings in their manuscript fully available?

Reviewer #1: Yes

Reviewer #2: Yes

4. Is the manuscript presented in an intelligible fashion and written in standard English?

Reviewer #1: Yes

Reviewer #2: Yes

5. Review Comments to the Author

Reviewer #1: This manuscript by Preston et al. provides several visualization scripts for the interpretation of dependent peptide (DP) search results in MaxQuant. Such post-analysis tools are currently lacking (at least I couldn't find any) and thus can be of a general interest to those performing MaxQuant DP searches.

Preston et al. provide original research results and describe new DP visualization tools that were tested on a generated NEM-treated BSA model dataset, as well as two public datasets that introduced in vitro protein modifications. Analyses are in general well-conducted and the applied methodology is well elaborated. Importantly, DP search results were adequately filtered as outlined in Table B in the supplementary text file. Also FDR statistics and protein localisations are provided by the popular, statistically sound MaxQuant suite. In case of the analysis of the NEM-treated BSA dataset and the public human blood proteome dataset, I believe the conclusion to follow after correct interpretation of the results. The raw proteomics data, peak lists and MaxQuant search results of the BSA dataset are provided in a public repository as required.

I have perhaps one major consideration where I am doubtful on the conclusion;

- In the presented results of the rat public dataset (results Fig. S8-S9), the frequency of these Δm is dramatically low if comparing to frequencies observed in the BSA model set. The modification of interest (+24.9606 Da) was identified solely once as a DP. So, the localization in Fig S9 would be then assumed on a single DP? Although given the a priori knowledge of this modification, this was considered to be a valid confirmation. However, it would most likely never be identified or distinguished without this knowledge a priori. I am unsure whether I am interpreting this correctly, or this is due to the quality of the data?

I have two further considerations that are I believe not critical points regarding criteria for publication but could improve the quality of the work:

1) A major aim of the manuscript is to promote the visualization tools and that proteomic researchers can apply these scripts in a user-friendly manner to interpret novel or anticipated modifications. Currently, there is a reference to an empty github user page (https://github.com/preston-gw/) where further development would be presented. Perhaps a good alternative to stimulate further usage is to gather these R scripts in a single R package that can be installed and loaded with a name that could be cited (instead of 'the visualization tools'). For instance, such R algorithms are often accessible via R/Bioconductor. Also scripts could be implemented as functions with an intuitive name, as currently (although well-elaborated) there is quite some complexity with five different script termed Script I to Script V. I don't believe this to be a critical point if viewing criteria for publication, although it could be I think greatly beneficial for future usage.

2) Overall the manuscript is focused on in vitro modifications, as stated in the title. However, the methodology could easily be applied on biological, in vivo modifications. For instance, it would be interesting to compare dependent peptide searches with or without a certain cellular stress or comparing wild-type versus mutants in PTM modifying enzymes. Certain scripts such as Script III would definitely be too stringent, but overall this is perhaps a missed chance to broaden up towards analysis of in vivo modifications that are of general interest to the proteomics community.

Besides these comments I noted several rather minor things during reading:

Abstract:

- Line 31-32: ‘To facilitate interpretation of the search results, of which there were hundreds, we ...’ , reads very vague. Are we talking here about identified dependent peptides or all identifications?

- Lines 34-36: after stating that public rat and human proteome data was searched, 19 expected modifications sites were anticipated. I’m confused, are we expecting these modifications in whole species proteomes or on the BSA model dataset that was generated? If on the model dataset, these anticipated modification sites would be best described before mentioning the whole proteome public datasets.

Introduction

- Line 49-50: I think an obvious reason being that specifying too many variable modifications will result in a search database expansion eventually having a negative effect on obtained peptide identifications at a certain FDR threshold.

- Line 64-65: ‘Mordret et al. recently carried out a validation of the function using phosphoproteome [12]’. I was initially confused as the cited reference focuses on amino acid substitutions. However they did assess the performance of the DP search using a model phosphorylation dataset. The sentence have to be rephrased to make f this more clear, it is in fact a very relevant work where the DP reliability was evaluated.

- Line 69-71: I guess an important incentive is that no such visualization tool is yet available for automated interpretation and visualization for MaxQuant dependent-peptide results?

Methods

- Line 94-95: The contaminant list from MaxQuant was edited to remove irrelevant or interfering sequences? I did not find any more information on this part.

-Lines 198: Script II removes DPs (peptide and Δm) identified in non-treated sample from the NEM-treated sample. Hence, in theory negative values can occur, but I guess these are not plotted or can we consider under-representation? Taken together, I feel that both Script II applies a rather simplistic rule to look at enrichment by simply subtracting DP peptides. Would it be more adequate considering for instance a more 'relative/normalized' approach, e.g. perhaps by considering a ratio of treated versus untreated or something similar?

FIGURES

- Fig S1: What is analysis 1 and 2, might be better termed as replicate 1 and 2?

- Fig S2: Is this indicative of any nucleophilic residue in trypsin that is known to be modified? Not specified in legend.

- Fig. 1:

o Irrespective of the frequency of certain Δm at a single localization, the rectangles would appear similar. Ideally, would it be possible to tune the transparency or a color gradient according to the frequency of Δm’s? Another way would be to look in certain mass bins of the Δm frequency plot and deduce whether any protein localization are overrepresented.

o Would it be worthwhile to indicate nucleophilic amino acids on the x-axis or with vertical lines if not making this too complex?

Overall, I believe this work to fulfill the publication criteria, though definitely further modifications could improve the manuscript.

Reviewer #2: The manuscript describes a tool containing several R scripts that aim to facilitate the analysis of data coming from a dependent peptide database search, looking for modified versions of identified proteins. The focus is on proper visualizations that give an overview about the found peptides and modifications.

The manuscript is well written but I still miss several items and am also not yet convinced about how much the visualization may help an interested user.

Major comments

1) The 3 presented figure types are quite simple and should be enhanced.

Taking figure 3, then the figure 3A could have thinner lines and maybe use transparency to further distinguish otherwise overlapping squares.

Also, as figures 3A and 3B are complementary, why not rotating the figure 3B 90 degrees and show it along with the y-axis of figure 3A?

I do not see that Figure 3C contains information about the mass shifts and thus it is limited in its usage, particularly for researchers that aim to understand a particular modification.

The use of colors is limited to some shades of blue and red lines. Given that we have the year 2020, I suggest to take advantage of a larger color palette.

How about zooming into regions of interest in a protein? Does the script allow to look into a region within the full protein sequence? And can one visualize a subset of the modifications, e.g. by specifying a smaller mass range?

When having a larger dataset with many proteins, is there any visualization of the full set of proteins, helping the researcher to find out which would be the proteins with particularly interesting patterns?

2) I miss the aim of the study. Which are scenarios where the tools will be of help? Does it target the investigation of samples containing one protein or simple mixtures, or also complex samples with thousands of proteins? If the latter was the case, what would the analysis procedure be?

3) What about the scripts' potential to visualize in vivo PTMs? This would be very interesting to a much broader readership. Another rather simple enhancement would be to look into the identify of the different mass shifts. They could be compared to UniMod/ResID databases and the putative names of the modifications could be extracted, and then for instance shown as word cloud.

Minor comments

- The introduction would benefit from a short comparison of open database search and dependent search, including their challenges and advantages

- I am missing a short description about the flexibility of the scripts to vary the visualizations and/or to extend them to include other information

6. PLOS authors have the option to publish the peer review history of their article (what does this mean?). If published, this will include your full peer review and any attached files.

Reviewer #1: No

Reviewer #2: No

---

## [Author Response · Author response to Decision Letter 0]

27 Apr 2020

PONE-D-20-02144

Response to reviewers

Academic Editor’s comments

As shown by the numerous questions of the reviewers the manuscript is currently not clear enough on the actual use of the visualisation tools that are proposed. The lack of details in the case studies and the subsequent confusion in understanding the visualisation is a concern. The revised version should not only include more detailed information but also refer to more of the existing resources describing PTMs.

Response: We thank the Academic Editor and the reviewers for their constructive comments. To address concerns about the use of the tools, we have revised the Introduction (final paragraph) and the Results and Discussion (‘Validation of tools using public data’, ‘Scope for extending the present study’, ‘Summary’). Specifically we have tried to better convey the idea that some scripts are primarily hypothesis-generating while others are hypothesis-testing. To address the lack of detail in the case studies, we have included extra information in Materials and Methods (‘Public data’) and Results and Discussion (‘Validation of tools using public data’ and ‘Summary’). We have also included a more detailed discussion of the effects of ‘fixed’ and ‘variable’ modifications on observed ∆m values (‘Visualisation of expected modifications’ in Results and Discussion). 

We have cited publications describing the Unimod and RESID databases (‘Scope for extending the present study’ in Results and Discussion) and have mentioned that Unimod was the source of mass data for some of our calculations (‘Mass calculations’ in Materials and Methods).

In addition, we have:

• clarified certain aspects of script development and usage (Materials and Methods);

• clarified certain aspects of figure preparation (‘Figure preparation’ in S1 Text);

• reworded a few phrases to make them clearer and more concise (e.g., in the Abstract);

• added detail to figure captions where necessary;

• cited an additional publication (ref. 21) in which dependent-peptide search results have been reported;

• pointed out that the results for haemoglobin δ-chain overlapped with those for haemoglobin β-chain (apologies – this did not come to light until the revision stage).

Further changes relating to reviewers’ comments are detailed below.

First reviewer’s comments

I have perhaps one major consideration where I am doubtful on the conclusion; - In the presented results of the rat public dataset (results Fig. S8-S9), the frequency of these Δm is dramatically low if comparing to frequencies observed in the BSA model set. The modification of interest (+24.9606 Da) was identified solely once as a DP. So, the localization in Fig S9 would be then assumed on a single DP? Although given the a priori knowledge of this modification, this was considered to be a valid confirmation. However, it would most likely never be identified or distinguished without this knowledge a priori. I am unsure whether I am interpreting this correctly, or this is due to the quality of the data?

Response: We acknowledge that the ∆m in question is not prominent in Figure S8, and therefore would not have been picked out using the simple survey approach. Also, the interpretation of this ∆m is not straightforward (it is actually the shift with respect to a pyridylethylated base peptide). Consider however that Figure S8 was generated using Script I, which cannot perform enrichment. If we been able to use Script II/III, then the ∆m of interest might have been more obvious. Still, Figure S8 does have some interesting features: the ∆m of +105 Da, for example, is likely due to pyridylethylation of non-cysteine residues. The sparsity of Figures S8 and S9 could reflect a real lack of modifications in the rat cofilin-1, or it could reflect a property of the sample or the quality of the data. Our finding of a single dependent peptide is actually consistent with Piroli and coauthors’ finding of a single peptide-spectrum match (see Table I in ref. 27). In the revised manuscript it has been made clear that Script I did not by itself identify the modification of interest. It has also been pointed out that Script III did not by itself identify expected modifications in plasma proteins. A related comment has been added to the ‘Summary’ section of Results and Discussion. A note regarding the 105-Da shift has been added to the caption of Figure S8.

A major aim of the manuscript is to promote the visualization tools and that proteomic researchers can apply these scripts in a user-friendly manner to interpret novel or anticipated modifications. Currently, there is a reference to an empty github user page (https://github.com/preston-gw/) where further development would be presented. Perhaps a good alternative to stimulate further usage is to gather these R scripts in a single R package that can be installed and loaded with a name that could be cited (instead of 'the visualization tools'). For instance, such R algorithms are often accessible via R/Bioconductor. Also scripts could be implemented as functions with an intuitive name, as currently (although well-elaborated) there is quite some complexity with five different script termed Script I to Script V. I don't believe this to be a critical point if viewing criteria for publication, although it could be I think greatly beneficial for future usage.

Response: We agree that combining the scripts in a package would promote their usability. Currently we do not have experience of preparing R packages (however we acknowledge that this could be gained without too much difficulty). We appreciate that a package would be easier to cite than five different numbered scripts. The five-script system does however establish a clear link between results and methods, and should aid reproducibility of results. The idea of integrating the scripts has been included in a new section of Results and Discussion (‘Scope for extending the present study’). A publication describing R packages and Bioconductor has been cited.

Overall the manuscript is focused on in vitro modifications, as stated in the title. However, the methodology could easily be applied on biological, in vivo modifications. For instance, it would be interesting to compare dependent peptide searches with or without a certain cellular stress or comparing wild-type versus mutants in PTM modifying enzymes. Certain scripts such as Script III would definitely be too stringent, but overall this is perhaps a missed chance to broaden up towards analysis of in vivo modifications that are of general interest to the proteomics community.

Response: We agree that tools for visualising in vivo modifications would be very useful to the proteomics community. The current scripts are probably more appropriate for looking at modifications not found in vivo. Scripts II, III and IV enrich dependent peptides for modifications that are unique to a test sample; they test for presence versus absence. For in vivo modifications, a more quantitative approach might be better. This idea has been included in the new section of Results and Discussion (‘Scope for extending the present study’).

Line 31-32: ‘To facilitate interpretation of the search results, of which there were hundreds, we ...’ , reads very vague. Are we talking here about identified dependent peptides or all identifications?

Response: Apologies – we meant dependent peptides only. The sentence has been revised.

Lines 34-36: after stating that public rat and human proteome data was searched, 19 expected modifications sites were anticipated. I’m confused, are we expecting these modifications in whole species proteomes or on the BSA model dataset that was generated? If on the model dataset, these anticipated modification sites would be best described before mentioning the whole proteome public datasets.

Response: Apologies – we meant expected sites in rat cofilin-1 and the human plasma proteins only. The sentence has been revised.

Line 49-50: I think an obvious reason being that specifying too many variable modifications will result in a search database expansion eventually having a negative effect on obtained peptide identifications at a certain FDR threshold.

Response: More detail has been added and a reference has been cited.

Line 64-65: ‘Mordret et al. recently carried out a validation of the function using phosphoproteome [12]’. I was initially confused as the cited reference focuses on amino acid substitutions. However they did assess the performance of the DP search using a model phosphorylation dataset. The sentence have to be rephrased to make f this more clear, it is in fact a very relevant work where the DP reliability was evaluated.

Response: We agree that this piece of supporting evidence is actually rather important. The description of Mordret and coauthors’ study has been expanded.

Line 69-71: I guess an important incentive is that no such visualization tool is yet available for automated interpretation and visualization for MaxQuant dependent-peptide results?

Response: Certainly we are not aware of any other tools that can map dependent peptides to sequences (although ∆m frequency histograms are quite common). The motivation for developing the tools has been stated in the Introduction. Also, we thought of two more references to cite: one describes MaxQuant’s viewer module, which can visualise raw mass spectrometry data and Andromeda search results; the other describes the Perseus software, which can draw histograms.

Line 94-95: The contaminant list from MaxQuant was edited to remove irrelevant or interfering sequences? I did not find any more information on this part.

Response: Further details have been added to S1 Text.

Lines 198: Script II removes DPs (peptide and Δm) identified in non-treated sample from the NEM-treated sample. Hence, in theory negative values can occur, but I guess these are not plotted or can we consider under-representation? Taken together, I feel that both Script II applies a rather simplistic rule to look at enrichment by simply subtracting DP peptides. Would it be more adequate considering for instance a more 'relative/normalized' approach, e.g. perhaps by considering a ratio of treated versus untreated or something similar?

Response: It is correct to say that the dependent peptides removed by Script II are not plotted (although one can get an idea of which ones they were by comparing Figures 1A and 1B). The negative values proposed by the reviewer would correspond to ∆m frequencies for the deleted peptides. In theory, Script II could be adapted to visualise these. We fully agree that a more quantitative approach would be helpful, particularly when dealing with in vivo modifications. The idea of quantitative comparisons has been included in the new section of Results and Discussion (‘Scope for extending the present study’).

Fig S1: What is analysis 1 and 2, might be better termed as replicate 1 and 2?

Response: Figures S1 and S3 have been updated. A reference to ‘analytical replicates’ has been made in Materials and Methods.

Fig S2: Is this indicative of any nucleophilic residue in trypsin that is known to be modified? Not specified in legend.

Response: None of these putative trypsin modifications could be attributed to a nucleophilic character of trypsin; they may all be ‘background’ modifications or artefacts. The main findings have summarised in the caption of Figure S2.

Fig. 1:

Irrespective of the frequency of certain Δm at a single localization, the rectangles would appear similar. Ideally, would it be possible to tune the transparency or a color gradient according to the frequency of Δm’s? Another way would be to look in certain mass bins of the Δm frequency plot and deduce whether any protein localization are overrepresented.

Response: Rectangles’ borders have been made partially transparent (Figures 1, 3, S2, S8 and S10-14). Scripts I-III have been updated (the transparency originates from the underlying code).

Would it be worthwhile to indicate nucleophilic amino acids on the x-axis or with vertical lines if not making this too complex?

Response: We agree that the annotation of nucleophilic residues could be very useful. We did try implementing the idea in early versions of Script IV but the resulting plots looked a bit cluttered. We also struggled to annotate disulfide bonds in a generalised way. The idea of annotating nucleophilic sites has been included in the new section (‘Scope for extending the present study’).

Second reviewer’s comments

The 3 presented figure types are quite simple and should be enhanced. Taking figure 3, then the figure 3A could have thinner lines and maybe use transparency to further distinguish otherwise overlapping squares.

Response: We agree. The line width has been reduced (Figures 1-3, S1 and S3) and rectangles’ borders have been made partially transparent (Figures 1, 3, S2, S8 and S10-14).

Also, as figures 3A and 3B are complementary, why not rotating the figure 3B 90 degrees and show it along with the y-axis of figure 3A?

Response: This is an excellent idea that had not occurred to us. Rotating a histogram is not straightforward in R, but could probably be achieved using ‘barplot’. The idea has been included in the new section of Results and Discussion (‘Scope for extending the present study’).

I do not see that Figure 3C contains information about the mass shifts and thus it is limited in its usage, particularly for researchers that aim to understand a particular modification.

Response: Figure 3C shows the result of a targeted data analysis using Script IV; it shows how the localisations probabilities for an expected mass shift (+390.10 ± 0.01 Da) are distributed within each dependent peptide. In the case of Figure 3C, the expected shift is given in the figure caption. For traceability, Script IV could be modified to print the expected shift on the plot, or to incorporate the shift into the name of an output file. Indeed, our usual approach was to incorporate the shift into the name of an *.svg file. The idea of stamping graphics with identifying information has been included in the new section of Results and Discussion (‘Scope for extending the present study’).

The use of colors is limited to some shades of blue and red lines. Given that we have the year 2020, I suggest to take advantage of a larger color palette.

Response: Adding transparency has generated some additional contrast in many of the figures, and we hope that this has gone some way to addressing the reviewer’s concerns. For consistency, the red colour has been added to all dependent-peptide localisation plots. For figures such as Figure 2, the chosen colour palette gets filtered through a partially-transparent blue strip. For this reason, we preferred to keep the colour palette as simple as possible (the blues in Figure 2 are actually greys as seen through a blue filter). The idea of a broader colour palette has been included in the new section of Results and Discussion (‘Scope for extending the present study’).

How about zooming into regions of interest in a protein? Does the script allow to look into a region within the full protein sequence? And can one visualize a subset of the modifications, e.g. by specifying a smaller mass range?

Response: To alter the dimensions of the graphics, users could either modify the underlying code or manually stretch the graphics window. Currently, the mass ranges are 1000 Da for the untargeted visualisations (e.g., Script III/Figure 3A) and 0.02 Da for the targeted visualisations (e.g., Script IV/Figure 3C). For targeted visualisations, the tolerance can be increased or decreased as required. Ways of adjusting the scripts have been mentioned in the new section of Results and Discussion (‘Scope for extending the present study’).

When having a larger dataset with many proteins, is there any visualization of the full set of proteins, helping the researcher to find out which would be the proteins with particularly interesting patterns?

Response: Scripts I-III generate one graphic per protein. Scripts IV and V generate one graphic per combination of protein and mass shift. In theory, these scripts could be modified to enumerate proteins, retrieve relevant protein sequences and generate batches of graphics. In fact, we did implement this idea albeit on a small scale: we were able to iterate Script III so that it cycled through a list of six plasma proteins; and to iterate Script IV so that it cycled through a list of expected modifications. The idea of iterating scripts has been included in the new section of Results and Discussion (‘Scope for extending the present study’). It has also been made clear that iteration was done in the present study.

I miss the aim of the study

Response: Our primary aim was to evaluate dependent peptide searching as a method for characterising modified proteins. Our secondary aim was to develop visualisation tools. The Abstract, Introduction and Summary have been revised so as to better convey the aims and structure of the study.

Which are scenarios where the tools will be of help?

Response: We envisage the tools as being useful for characterising proteins that have been exposed to enzyme activities or reactive small molecules. If modifications are unique to a given test sample, then the more advanced tools may be able to pick them out. Possible applications have been mentioned at the end of the introduction. The idea that some of the tools are hypothesis-generating while others are hypothesis-testing has also been mentioned.

Does it target the investigation of samples containing one protein or simple mixtures, or also complex samples with thousands of proteins? If the latter was the case, what would the analysis procedure be?

Response: Each script targets a single protein but in theory could be iterated to cycle through a list of proteins (see earlier answer). In practice, this would require a way of cataloguing multiple graphics. An analytical procedure has been proposed in the new section of Results and Discussion (‘Scope for extending the present study’).

What about the scripts' potential to visualize in vivo PTMs? This would be very interesting to a much broader readership.

Response: We agree that the visualisation of in vivo PTMs would be an impactful next step. The idea has been mentioned in the new section of Results and Discussion (‘Scope for extending the present study’).

Another rather simple enhancement would be to look into the identify of the different mass shifts. They could be compared to UniMod/ResID databases and the putative names of the modifications could be extracted, and then for instance shown as word cloud.

Response: Automating the identification in this way could be very powerful, but direct annotation of the plots would have to be done carefully so as not to obscure data. We like the idea of the word cloud. Ways of annotating the visualisations have been proposed in the new section of Results and Discussion (‘Scope for extending the present study’). Publications describing Unimod and RESID have been cited.

The introduction would benefit from a short comparison of open database search and dependent search, including their challenges and advantages

Response: The difference between open database searches and spectral pair methods has been explained. The advantages and limitations of dependent peptide searching have been made clear.

I am missing a short description about the flexibility of the scripts to vary the visualizations and/or to extend them to include other information.

Response: We hope that the new section of Results and Discussion (‘Scope for extending the present study’) addresses the reviewer’s concern.

---

## [Decision Letter · Decision Letter 1]

12 Jun 2020

Visualisation tools for dependent peptide searches to support the exploration of *in vitro* protein modifications

PONE-D-20-02144R1

Dear Dr. Preston,

We’re pleased to inform you that your manuscript has been judged scientifically suitable for publication and will be formally accepted for publication once it meets all outstanding technical requirements.

Kind regards,

Frederique Lisacek

Academic Editor

PLOS ONE

Additional Editor Comments (optional):

Reviewers' comments:

Reviewer's Responses to Questions

**Comments to the Author**

1. If the authors have adequately addressed your comments raised in a previous round of review and you feel that this manuscript is now acceptable for publication, you may indicate that here to bypass the “Comments to the Author” section, enter your conflict of interest statement in the “Confidential to Editor” section, and submit your "Accept" recommendation.

Reviewer #1: All comments have been addressed

2. Is the manuscript technically sound, and do the data support the conclusions?

Reviewer #1: Yes

3. Has the statistical analysis been performed appropriately and rigorously? 

Reviewer #1: N/A

4. Have the authors made all data underlying the findings in their manuscript fully available?

Reviewer #1: Yes

5. Is the manuscript presented in an intelligible fashion and written in standard English?

Reviewer #1: Yes

6. Review Comments to the Author

Reviewer #1: Overall, I am satisfied with the changes made, which improve the quality of the work. I believe the scripts indeed can be useful for visualizing potential modifications for a specific protein of interest. However, as outlined by the new section 'Scope for extending the present study', still several improvements could be made for visualisation and comprehensive analysis of DP results. Taken together, this manuscript could form an incentive towards development of tools performing wider interpretation and visualization of DP results (e.g. full proteomes) - as exist actually for open searches.

7. PLOS authors have the option to publish the peer review history of their article (what does this mean?). If published, this will include your full peer review and any attached files.

Reviewer #1: No

---

## [Editor Report · Acceptance letter]

26 Jun 2020

PONE-D-20-02144R1 

Visualisation tools for dependent peptide searches to support the exploration of *in vitro* protein modifications 

Dear Dr. Preston:

I'm pleased to inform you that your manuscript has been deemed suitable for publication in PLOS ONE. Congratulations! Your manuscript is now with our production department. 

Kind regards, 

on behalf of

Dr. Frederique Lisacek 

Academic Editor

PLOS ONE